# Responses to the Tepotinib in Gastric Cancers with MET Amplification or MET Exon 14 Skipping Mutations and High Expression of Both PD-L1 and CD44

**DOI:** 10.3390/cancers14143444

**Published:** 2022-07-15

**Authors:** Sung-Hwa Sohn, Hee Jung Sul, Bum Jun Kim, Dae Young Zang

**Affiliations:** 1Hallym Translational Research Institute, Hallym University Sacred Heart Hospital, Anyang-si 14066, Korea; iisupy@korea.ac.kr (S.-H.S.); glwjd82@naver.com (H.J.S.); 2Department of Internal Medicine, Hallym University Sacred Heart Hospital, Hallym University Medical Center, Hallym University College of Medicine, Anyang-si 14068, Korea; getwisdom1025@gmail.com

**Keywords:** c-MET, gastric cancer, tepotinib

## Abstract

**Simple Summary:**

High MET copy-number variation (CNV) and high MET protein expression via MET exon 14 skipping mutation (METex14SM) are associated with tumorigenesis; however, METex14SM is rare in GC, so target therapy is rarely applicable. In this study, we analyzed GC panels using targeted next-generation sequencing in GC cell lines and found that tepotinib demonstrated therapeutic effects in MET-amplified GC and those with high expressions of PD-L1, CD44, and METex14SM. Our in vitro findings indicate that tepotinib prevents GC-associated MET amplification and MET mutation.

**Abstract:**

Both MET exon 14 skipping mutation (METex14SM) and high copy-number variation (CNV) lead to enhanced carcinogenesis; additionally, programmed-death ligand 1 (PD-L1) is often upregulated in cancers. In this study, we characterized the expression of MET (including METex14SM), PD-L1, and CD44 in human gastric cancer (GC) cells as well as the differential susceptibility of these cells to tepotinib. Tepotinib treatments inhibited the growth of five GC cells in a dose-dependent manner with a concomitant induction of cell death. Tepotinib treatments also significantly reduced the expression of phospho-MET, total MET, c-Myc, VEGFR2, and Snail protein in SNU620, MKN45, and Hs746T cells. Notably, tepotinib significantly reduced the expression of CD44 and PD-L1 in METex14SM Hs746T cells. By contrast, tepotinib was only slightly active against SNU638 and KATO III cells. Migration was reduced to a greater extent in the tepotinib-treated group than in the control group. Tepotinib may have therapeutic effects on c-MET-amplified GC, a high expression of both PD-L1 and CD44, and METex14SM. Clinical studies are needed to confirm these therapeutic effects.

## 1. Introduction

Three receptor tyrosine kinases (RTKs), MET, ErbB2, and FGFR2, have been widely studied in gastric cancer (GC). Among these, MET is overexpressed in advanced GC, regardless of tumor differentiation [1]. The aberrant activation of MET suppresses apoptosis and promotes cell proliferation, survival, migration, and angiogenesis as a result of various genetic alterations including gene amplification, overexpression, and mutation [2,3,4,5]. Only 7% of advanced GC patients show MET overexpression, although MET amplification occurs in 2–20% of GC patients [6,7]. MET exon 14 (METex14) alteration occurs with low frequency in stomach (4.8–7.1%), colorectal (~0–9.3%), and lung cancers (adenocarcinoma, 2.6–3.2%; pulmonary sarcomatoid tumor, 2.6–31.8%; adenosquamous carcinoma, 4–8.2%) [8,9,10]. METex14-altered lung cancer has been identified in patients expressing PD-L1 (*n* = 147; PD-L1 expression, 0%, 1–49%, and 50% accounting for 37%, 22%, and 41%, respectively, across 111 evaluable tumor samples); however, neither interaction nor complex formation was detected between c-MET and PD-L1 [11]. The tumorigenicity of CD44 is driven partly by the promotion of PD-L1 expression [12], which may mediate the chemoresistance of breast, lung, and prostate cancers [12,13].

Precision medicine is important in the management of GC. Currently, MET inhibitors have been approved for patients with GC and MET amplification (defined as a copy number increase) [14,15], overexpression (defined as increased mRNA expression), fusion, mutation, or rearrangement. With the improved efficacy of targeted therapies, comprehensive tumor profiling is needed. Somatic and inheritance-acquired copy-number variations (CNVs) in the genome have been found to be associated with cancer [16,17,18]. In particular, high CNVs (i.e., gene amplification) affect tumorigenesis in many types of cancer such as GC and colon, liver, and lung cancers [19,20]. The clinical use of next-generation sequencing (NGS) panels allows for the simultaneous assessment of targeted genes and entire genomes using a limited quantity of GC samples [21]. Cell-line models of target gene amplification (similar to gene mutation) or GC overexpression have consistently matched clinical responses showing susceptibility to targeted gene inhibitors.

Here, we characterized the expression of MET (including METex14SM and CNV), PD-L1, and CD44 in five GC cells using targeted NGS gene panel data and immunoblotting and further evaluated the differential susceptibility of the MET inhibitor tepotinib based on dose-dependent viability, cell death, and migration.

## 2. Materials and Methods

### 2.1. Cell Culture and Drug Treatment

Hs746T, SNU620, MKN45, KATO III, and SNU638 cell lines were purchased from KCLB (Korean Cell Line Bank, Seoul, Korea). Hs746T cells were maintained in either Dulbecco’s Modified Eagle Medium (DMEM; Thermo Fisher Scientific, Waltham, MA, USA), and all other cell lines were maintained in RPMI1640 and supplemented with 10% fetal bovine serum and 1% penicillin/streptomycin. The cells were cultured using standard procedures. MET-inhibitor tepotinib (EMD 1214063) was obtained from Selleck Chemicals (Houston, TX, USA).

### 2.2. GC Panel 

The GC panel is designed to assess clinically relevant mutations in 286 genes associated with carcinogenesis risk through the detection of CNV, insertion/deletions (indels), and single nucleotide variants (SNVs) located in the DNA coding sequences of the targeted genes (Table 1). We added a tiling probe in clinically actionable amplification candidate genes including CCND1, CCND2, CCND3, CD274, CDKN2A, EGFR, ERBB2, ERBB3, FGFR2, HGF, and MET. In addition, we added microsatellite (MSI) markers including BAT-25, BAT-26, and NR-24. The GC panel detected fusion genes including ALK, ROS, RET, EWSR1, and TMPRSS2.

### 2.3. Target Sequencing and Analysis

Target capturing sequencing was conducted using a customized target kit (Agilent Technologies, Santa Clara, CA, USA) according to the protocol provided by the manufacturer. DNA libraries were constructed according to the manufacturer’s instructions, and the customized target kit was performed using the Illumina NovaSeq6000 platform (Illumina, San Diego, CA, USA) to generate paired-end reads (150 bp). We used the cutadapt and sickle tools (v1.8.1, available for download at https://github.com/najoshi/sickle) to remove adapter sequences and low-quality sequence reads. The Burrows–Wheeler aligner [22] was used to align the sequencing reads onto the human reference genome (hg19). We used the MuTect algorithm in the Genome Analysis Tool Kit (GATK) [23] for score recalibration, local realignment, and filtering of sequence data. The Picard (v1.92, available for download at http://broadinstitute.github.io/picard) and Samtools programs [24] were used for basic processing and management of the sequencing data, and to generate mpileup files. To call variants, we used the VarScan v2.3.9 program with the mpileup2indel and mpileup2snp subcommands. SnpEff v4.2 software [25] was used to select variants located in coding sequences and predict their functional consequences (e.g., silent vs. non-silent variants). We used the CNVKit v0.8.5 program [26] to detect CNV. The PureCN program [27] was used to estimate tumor purity, ploidy, copy number, and the loss of heterozygosity.

### 2.4. Growth Inhibition Assays 

Half maximal inhibitory concentration (IC50) values for tepotinib in Hs746T, SNU620, MKN45, KATO III, and SNU638 cell lines were measured using an MTS assay for tepotinib concentrations of 20, 10, 1, 0.1, 0.01, 0.001, 0.0001, 0.00001, and 0.000001 µM for 48 h. The MTS assay was conducted according to our previously described method [28].

### 2.5. Cell Death Analysis 

Hs746T, SNU620, MKN45, KATO III, and SNU638 cell lines were seeded into 6-well plates at a density of 5 × 10^4^ cells/mL, and then treated with 10 nM or 1 µM of tepotinib. Cell apoptosis and necrosis were examined according to our previously described method [29]. 

### 2.6. Cell Migration Analysis

The analysis method for cell migration has been reported previously in detail [29]. After incubating for 48–72 h, cell migration was photographed and compared with the initial migration at 0 h.

### 2.7. Quantitative Real-Time (qRT) PCR Analysis

qRT PCR analysis was completed according to our previously described method [30]. Transcript levels of GAPDH were used for sample normalization. The primer sequences were as follows: COX-2 (F: TGA GCA TCT ACG GTT TGC TG-3′; R: AAC TGC TCA TCA CCC CAT TC-3′), GSK3β (F: GAA CTC CAA CAA GGG AGC AA-3′; R: GGG TCG GAA GAC CTT AGT CC-3′), MET (F: AAG AGG GCA TTT TGG TTG TG-3′; R: GAT GAT TCC CTC GGT CAG AA-3′), CCND1 (F: GAT CAA GTG TGA CCC GGA CT-3′; R: TCC TCC TCT TCC TCC TCC TC-3′), and GAPDH (F: TTC ACC ACC ATG GAG AAG GC-3′; R: GGC ATG GAC TGT GGT CAT GA-3′).

### 2.8. Western Blot Analysis

Western blot analysis was performed using standard procedures. Commercially available primary antibodies were directed against anti-MET (#4560; 1:1000; Cell Signaling Technology (CST), Danvers, MA, USA), anti-phospho-MET (#3077; 1:1000; CST), anti-SNAIL (#3879; 1:1000; CST), anti-β-catenin (#610153; 1:1000; BD Biosciences, Franklin Lake, NJ, USA), anti-VEGFR2 (#9698; 1:1000; CST), anti-PD-L1 (#13684; 1:1000; CST), anti-CD44 (#3570; 1:1000; CST), anti-c-MYC (sc40; 1:1000; Santa Cruz Biotechnology, Dallas, TX, USA), and anti-GAPDH (sc32233; 1:4000; Santa Cruz Biotechnology).

### 2.9. Statistical Analyses

IC_50_ values were calculated using nonlinear regression analysis. The percentages of intact and dead cells (apoptotic and necrotic) were calculated using the CytExpert software (Beckman Coulter). All data were analyzed using the Prism 5 software (GraphPad Software Inc., San Diego, CA, USA). Data are means ± standard deviation (SD). Statistical significance was examined by one-way analysis of variance (ANOVA). A *p*-value of < 0.05 was considered indicative of statistical significance.

## 3. Results

### 3.1. GC Cell Characteristics

GC panels of five GC cells (Hs746T, SNU638, KATO III, MKN45, and SNU620) were analyzed using targeted sequencing at Theragen Bio Institute, Seongnam, Korea. The 286 GC panel genes and their individual genetic aberrations are shown in Figure 1 and Appendix A. Further information on GC cell, including concomitant pathogenic variants and genetic aberrations, is listed in Table 1 and Table 2. A pathogenic mutation (CYP2D6) was detected in all GC cell lines (Table 1 and Appendix A). Among the 286 GC panel genes, we identified 21 amplified genes (≥5.0 copies) (Table 2); among these, nearly all GC cells were MET-amplified, except for the SNU638 cell line.

### 3.2. Effects of Tepotinib on Cell Viability in GC Cell Lines with or without MET and PD-L1 Expression

We tested the dose dependency of tepotinib inhibitory effects in SNU620, MKN45, Hs746T, Kato III, and SNU638 cells (Figure 2). The MET gene structure and copy number results are shown in Figure 2A; MET is encoded by 21 exons. SNU620, MKN45, and Hs746T cells had ≥30 copies and displayed high p-MET and MET protein expression, whereas ≤7 copies were found in Kato III and SNU638 cells and p-MET protein expression was not detected (ND) or low (Figure 2A,C,D and Appendix A). Notably, Hs746T cells displayed exon 14 skipping mutation low MET mRNA expression, and high p-MET, MET, CD44, and PD-L1 protein expression (Figure 2A–D). The discordance between low MET mRNA expression and high MET protein expression was caused by MET exon 14 deletion, which inhibited cbi binding and led to prolonged MET protein stability extended cell signaling on ligand stimulation, and ultimately increased MET activation and tumorigenicity [31,32,33]. Treatment with tepotinib decreased cell viability in all GC cells in a dose-dependent manner (Figure 2E). Non-linear regression analysis displayed tepotinib IC_50_ values of 6.2 nM for SNU620, 36.7 nM for MKN45, 4.4 µM for Hs746T, 6.9 µM for SNU638, and 7.9 µM for Kato III cells (Figure 2E).

### 3.3. Effects of Tepotinib on Cell Death

To investigate the effects of tepotinib on cell death in SNU620, MKN45, Hs746T, Kato III, and SNU638 cell lines, we examined apoptosis and necrosis using flow cytometry (Figure 3). Tepotinib demonstrated high apoptosis and necrosis rates in SNU620, Hs746T, and MKN45 cell lines, whereas cell death rates were very low in Kato III and SNU638 cells, occurring in a dose-dependent manner following tepotinib exposure for 48 h (Figure 3). SNU620, MKN45, and Hs746t cells presented high-MET CNV, whereas others such as Kato III and SNU638 presented low-MET CNV. Notably, Hs746T cells formed a high-CD44, PD-L1 expressor subtype (Figure 2C).

### 3.4. Effects of Tepotinib on Expression of Gene and Protein in GC Cell Lines

We measured the levels of carcinogenesis- and inflammation-related genes such as MET, GSK3β, CCND1, and COX2 to determine the effects of tepotinib in GC cells with MET amplification or a high expression of both PD-L1 and METex14SM. Following treatments with tepotinib, CCND1 and COX2 mRNA levels decreased in Hs746T and MKN45 cell lines, whereas GSK3β expression increased. By contrast, tepotinib demonstrated little effects in Kato III cells (Figure 4).

Immunoblot analysis revealed a downregulation of phosphor-MET, MET, VEGFR2, c-MYC, and Snail in tepotinib-treated GC cell lines, with the exception of MET-negative Kato III cells (Figure 5 and Appendix A). Notably, CD44 and PDL1 proteins decreased in tepotinib-treated Hs746T cells.

### 3.5. Effects of Tepotinib on Cell Migration in GC Cell Lines

To evaluate the inhibitory effect of tepotinib on the migration ability of MKN45 (high MET CNV), Hs746T (high MET CNV, METex14SM, high CD44, and high PD-L1 protein expression), and SNU638 (ND MET CNV and high PD-L1 protein expression) cell lines, we investigated a migration assay of these cell lines (Figure 6A). After 72 h, the MKN45 control cell line filled 55.17% of the wounded area, and the tepotinib-treated MKN45 cell line filled 4.15% of the wounded area (Figure 6B). After 48 h, the Hs746T and SNU638 control cell lines filled 100% and 90.36% of the wounded area, respectively, whereas tepotinib-treated Hs746T and SNU638 cell lines filled 35.53% and 55.67% of the wounded area, respectively (Figure 6B).

## 4. Discussion

Precision cancer medicine has developed rapidly with the advent of high-throughput NGS, and become an important component in personalized cancer therapy management. A significant proportion of NGS technologies are driven by CNV and structural variation (fusion, mutation, or rearrangement) across the genome. NGS-based multigene panel analysis provides information on the targeted genes that are susceptible to cancer [34,35]. Cancer cell lines are useful for the development or selection of targeted agents based on cell characteristics identified by NGS-based multigene panels. MET has received considerable attention as a potential target for GC therapy. In GC, the gain-of-function of MET is associated with promoting cancer cell survival, migration, and angiogenesis through MET amplification [4,36]. Therefore, in this study, we investigated the characteristics of five GC cell lines using 286 NGS cancer panels for therapeutic target selection and drug efficacy testing.

Among 286 GC panel genes, five GC cells had no mutation on AMER1, ARFRP1, B2M, BAP1, BCL2L1, BRD4, CBFB, CDK4, CDKN2B, CTNNB1, FGF19, FGF4, GATA1, IGF2, INHBA, KAT6A, KMT2A, KMT2C, KMT2D, MYCN, PTK2, and VHL. The five GC cells had pathogenic variants (CYP2D6 missense mutation). Among these, MKN45 and SNU638 cells had drug response-related variants (CYP2C19 synonymous mutation or stop gain). CYP2C19 is a principal enzyme involved in the metabolism of clinically important drugs such as β-adrenoceptor blockers, antiulcer medications, anticonvulsant drugs, and antidepressants [37,38]. Importantly, CYP2C19 loss-of-function alleles are associated with adverse cardiovascular events, including stent thrombosis, stroke, and myocardial infarction caused by reduced responsiveness to the antiplatelet drug clopidogrel [39,40]. In this study, all five GC cells displayed MET expression, whereas Kato III cells indicated no p-MET expression. We tested the expression of CD44 and PD-L1 in the five GC cells and found that both Hs746T and SNU638 displayed PD-L1 expression, whereas only Hs746T displayed CD44 expression. This finding suggests that Hs746T and SNU638 may possess chemoresistance through PD-L1 or CD44 expression; therefore, Hs746T and SNU638 had higher tepotinib IC50 values than other cell lines. However, SNU620, MKN45, and Hs746t cells had CNV ≥ 30 for MET and higher cell death rates than Kato III and SNU638 cells. Interestingly, despite Kato III having CNV = 7 for MET, this cell line had a higher tepotinib IC50 value and lower cell death rate than other cell lines (FGFR2, CNV = 87; CTNNB1, CNV = 22). This GC cell information may be used as a preclinical model system for drug screening, which could prove useful in the development of new drugs.

The Wnt/β-catenin and MET signaling pathways are associated with GC progression events. The snail enhances Wnt/β-catenin pathway activation by interacting with β-catenin [41]. The Wnt/β-catenin signaling pathway regulates the expression of the canonical cancer stem cell marker CD44 directly or through the intermediate c-Myc [42,43]. CD44 also strongly enhances MET and KDR (VEGFR2) signaling [44,45], and collaboration between Wnt/β-catenin, CD44, and MET can be an important factor in tumorigenesis. CD44 may also counteract programmed cell death, which leads to tumorigenesis [46] and partly promotes PD-L1 expression to mediate cancer cell proliferation and immune evasion [12]. CD44 and PD-L1 expression may partly contribute to the tumorigenic, immunosuppressive, and chemoresistant traits of cancers [13]. Our results indicate that tepotinib may suppress β-catenin, CD44, PD-L1, c-MYC, KDR, MET, and p-MET through increased GSK3β expression. Tepotinib also inhibited GC cell migration.

## 5. Conclusions

In this study, we could show that the combined blockading of MET, CD44, and PD-L1 improved the GC therapeutic efficacy of tepotinib. Our results strongly support the clinical evaluation of tepotinib, which prevents GC association with CD44, PD-L1, and c-MET.

## Figures and Tables

**Figure 1 cancers-14-03444-f001:**
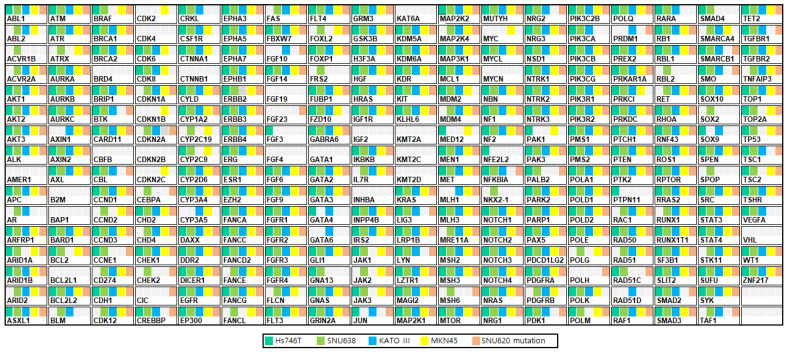
Gene mutations in human gastric cancer cells measured using a targeted next-generation sequencing (NGS) panel. Excel was used to generate Figure 1.

**Figure 2 cancers-14-03444-f002:**
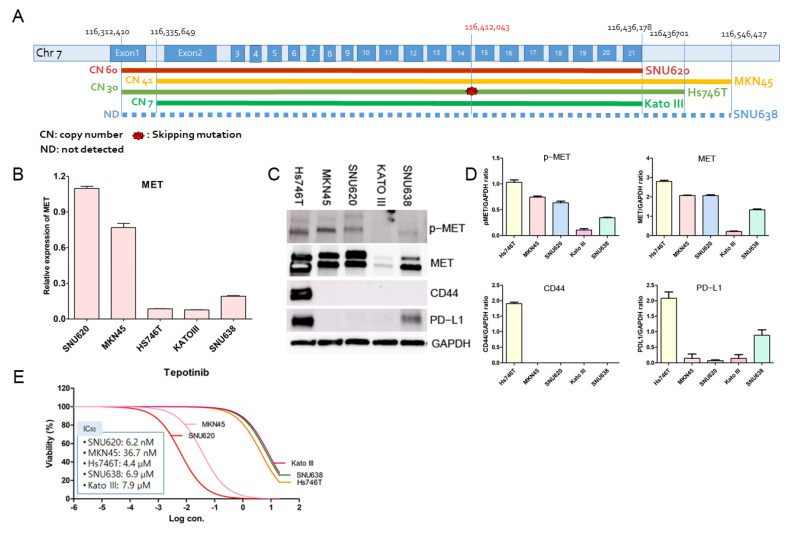
Effects of tepotinib on cell viability in GC cell lines with or without MET gene overexpression and p-MET, MET, CD44, and PD-L1 protein expression. (**A**) Human GC cell lines, MET gene structures, and copy numbers (CNs) are measured by targeted NGS. (**B**) MET mRNA expression levels were determined by qRT-PCR. (**C**) Protein expression levels of p-MET, MET, CD44, and PD-L1 were determined using Western blot analysis. (**D**) Using the image J for densitometry analysis. (**E**) Five GC cell lines were treated with various concentrations of tepotinib for 48 h.

**Figure 3 cancers-14-03444-f003:**
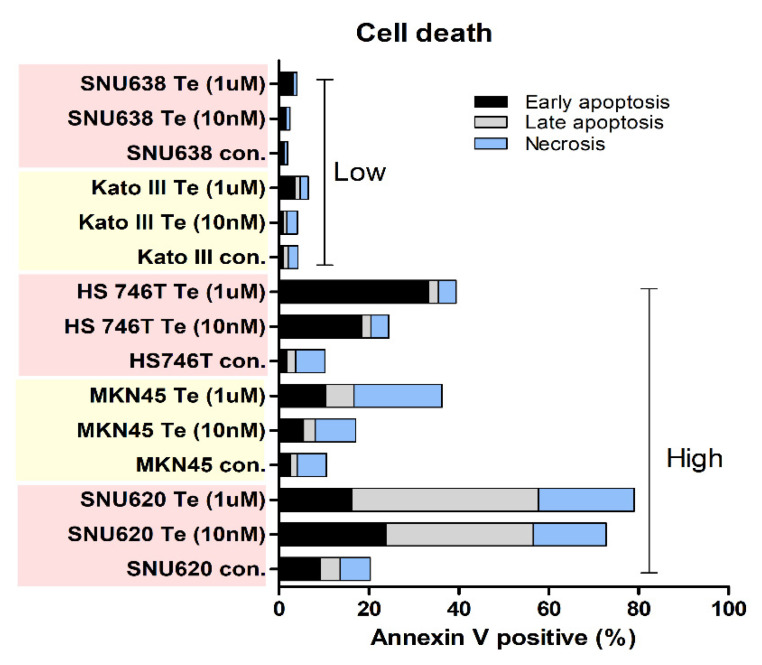
Effects of tepotinib on cell apoptosis and necrosis in GC cell lines. SNU638, Kato III, Hs746T, MKN45, and SNU620 cell lines were treated with 10 nM or 1 µM of tepotinib for 48 h.

**Figure 4 cancers-14-03444-f004:**
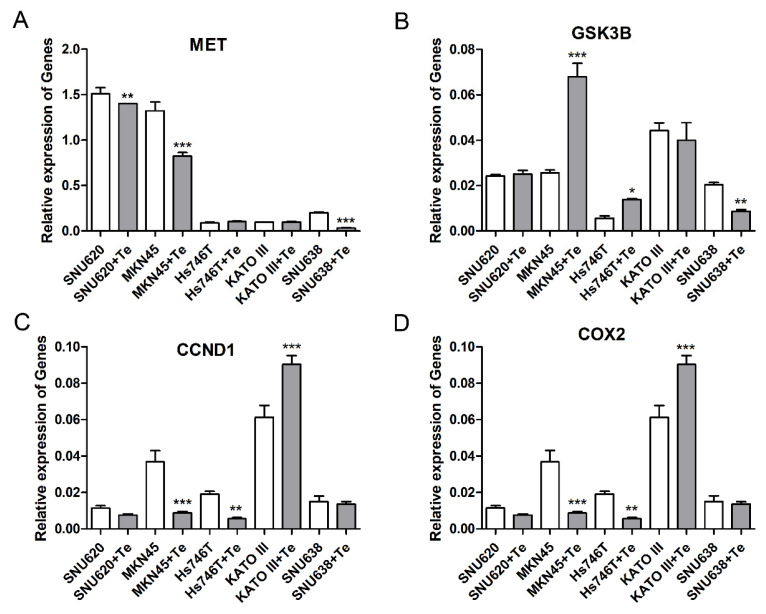
Effects of tepotinib on gene expressions of (**A**) MET, (**B**) GSK3β, (**C**) CCND1, and (**D**) COX2 in GC cells. Gene expressions were determined by qRT-PCR. MKN45, Hs746T, Kato III, and SNU638 cell lines were treated with 1 µM tepotinib, and SNU620 cells were treated with 10 nM tepotinib for 48 h. *, *p* < 0.05; **, *p* < 0.01; and ***, *p* < 0.001 compared with the control group.

**Figure 5 cancers-14-03444-f005:**
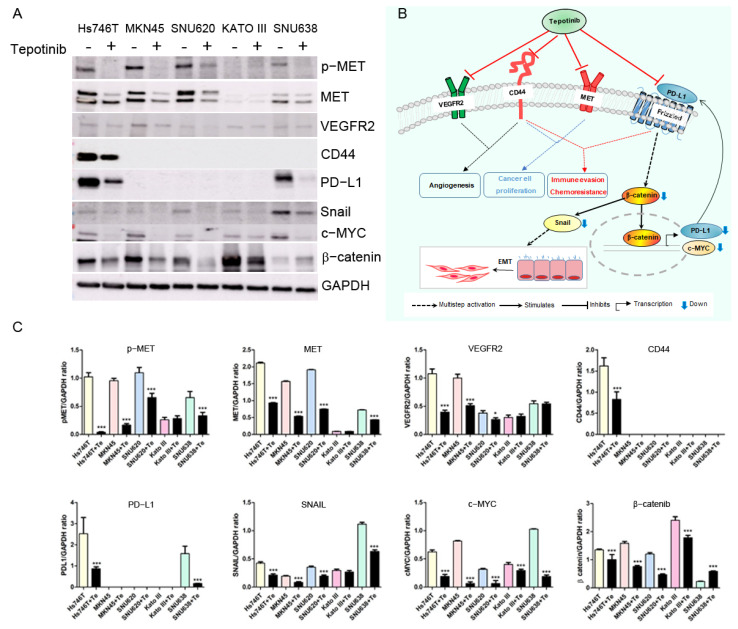
Effects of tepotinib on protein expression in GC cells. Protein expression of SNU620, MKN45, Hs746T, Kato III, and SNU638 cell lines. (**A**) Protein expression levels of MET, p-MET, VEGFR2, CD44, PDL1, Snail, c-MYC, and β-catenin were determined using immunoblot analysis. Hs746T, MKN45, Kato III, and SNU638 cell lines were treated with 1 µM tepotinib, and the SNU620 cell line was treated with 10 nM tepotinib for 48 h. (**B**) Graphic overview of tepotinib-treated interventions in GCs with MET amplification or METex14SM and high expression of both PD-L1 and CD44. (**C**) Using the image J for densitometry analysis. * *p* < 0.05 and *** *p* < 0.001 compared with the control group.

**Figure 6 cancers-14-03444-f006:**
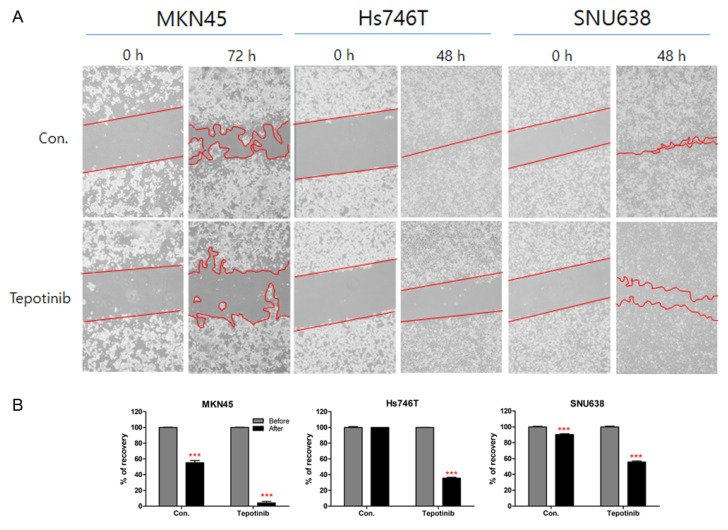
Effects of tepotinib on cell migration in GC cells. (**A**) A wound-healing assay was used to assess the effect of tepotinib on the migration ability of MKN45, Hs746T, and SNU638 cell lines. Hs746T, MKN45, and SNU638 cell lines were treated with 1 µM tepotinib for 48–72 h. (**B**) Using the image J for wounded area analysis. Con, non-treated control cells. *** *p* < 0.001 compared with the control group.

**Table 1 cancers-14-03444-t001:** Summary of pathogenic variants of individual cells.

Cell Line	Gene	DNA Change	Protein Change	Clinical Effect	Exonic Effect
SNU620	CYP2D6	c.886T>C	p.Cys296Arg	Pathogenic	Missense variant
		c.733T>C	p.Cys 245Arg	Pathogenic	Missense variant
MKN45	CYP2C19	c.681G>A	p.Pro227Pro	drug response	synonymous_variant
	CYP2D6	c.886T>C	p.Cys296Arg	pathogenic	missense_variant
		c.733T>C	p.Cys 245Arg	pathogenic	missense_variant
Hs746T	CYP2D6	c.886T>C	p.Cys296Arg	pathogenic	missense_variant
		c.733T>C	p.Cys 245Arg	pathogenic	missense_variant
KATO III	CYP2D6	c.1457C>G	p.Thr486Ser	pathogenic	missense_variant
		c.1304C>G	p.Thr 435Ser	pathogenic	missense_variant
		c.886T>C	p.Cys296Arg	pathogenic	missense_variant
		c.733T>C	p.Cys 245Arg	pathogenic	missense_variant
SNU638	CYP2C19	c.636G>A	p.Trp212 *	drug response	stop_gained
		c.681G>A	p.Pro227Pro	drug response	synonymous_variant
	CYP2D6	c.1457C>G	p.Thr486Ser	pathogenic	missense_variant
		c.1304C>G	p.Thr 435Ser	pathogenic	missense_variant
		c.886T>C	p.Cys296Arg	pathogenic	missense_variant
		c.733T>C	p.Cys 245Arg	pathogenic	missense_variant

**Table 2 cancers-14-03444-t002:** Description and concomitant genomic alteration of individual cells.

Cell Line	MET CNV	Other Genes (CNV)	SNP	INS	DEL	Silent Mutation	Missense Mutation	Nonsense Mutation
SNU620	60	CCND3 (32), CYP2C19 (5), CYP2C9 (5)	15,340	148	167	471	304	1
MKN45	41	CYP2C19 (6), CYP2C9 (6), SLIT2 (5), MDM2 (5), FRS2 (5), POLA1 (5)	13,359	124	165	403	246	2
Hs746T	30	CYP3A4 (8), CYP3A5 (8), CCND1 (8), PIK3CG (7), SYK (6), FANCC (6), PTCH1 (6), TGFBR2 (5), NRG3 (5), PTEN (5), FAS (5), CYP2C19 (5), CYP2C9 (5)	13,366	134	155	377	254	16
Kato III	7	FGFR2 (87), CTNNB1 (22)	14,995	134	184	431	323	4
SNU638	ND	ND	15,231	150	603	551	335	4

ND, not detected; CNV, copy number variance; SNP, single-nucleotide polymorphism; INS, insertion mutation; DEL, deletion mutation.

## Data Availability

All data presented in this study are available in the main body of the manuscript. Data generated for the current study are available from the corresponding author upon reasonable request.

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
