# Peer review of "Responses to the Tepotinib in Gastric Cancers with MET Amplification or MET Exon 14 Skipping Mutations and High Expression of Both PD-L1 and CD44"

_cancers, 2022, doi:10.3390/cancers14143444_

Round 1

Reviewer 1 Report

This is a sound paper evaluating the in vitro responses of a MET inhibitor in gastric cancer cell lines. The techniques are conventional and well accepted and the results are encouraging for future in vivo studies.

The introduction is rather sparse. More background on MET function in cancer cells and its roles in PDL1 and CD44 signaling would be very helpful.  I would suggest a figure outlining the intracellular interactions between theses proteins, showing where in these pathways these inhibitors work and what the downstream effects are.

Author Response

We agree with reviewer’s concern. We added the pathways in figure 5B.

Reviewer 2 Report

Thank you to the authors on their submission. The manuscript lacks cohesiveness and there are concerns regarding the design of the study. E.g.: Why was a cell line that has low MET expression still chosen for met inhibition? Reasoning not provided.  How the doses of 10 and 1uM of tepotinib were chosen for the cell death analysis?  

Author Response

When using met inhibitors in clinical practice, the cutoff value of MET inhibitors is important. So, to check the cutoff value of MET inhibitor, we tested low MET expression cells together.

The ic50 value of SNU620 is 6.2 nM, but other cells (Hs746T, SNU638, and Kato III) are more than 4 uM. However, in SNU620, more than 80% of cells died at 1 uM or higher, so we could not compare the experimental results with other cells, so we compared cell death at 10 nM and 1 uM.

Reviewer 3 Report

This is an interesting study that expression of MET, PD-L1, and CD44 in gastric cancer cell lines. The study evaluated the differential susceptibility of c-MET inhibitor tepotinib on cell viability, apoptosis and migration. The authors identified pathogenic mutation CYP2D6 in all cell lines as well as amplified genes including MET-amplification. The study identifies that treatment with tepotinib downregulates CCND1 and COX2 and upregulates GSK3β expression. Downregulation of phosphor-MET, MET, VEGFR2, Snail, and c-MYC was also observed in tepotinib-treated cells. The work is original, methods are provided in detail and the manuscript is well-written without typographical errors.

Comments

The manuscript should be updated to include some essential details on variant calling and filtering.

1-    Please include more information on variants allele frequency in the gnomAD and ExAC databases.

2-    Add more information on read depth and the total depth for calling variants at a specific locus.

3-    Details on filtering out synonymous mutations and allele frequency threshold for filtering non-synonymous mutations.

4-     How were raw variants filtered?

5-    Are any of the variants present in the 1000 Genomes? This is required to make sure only somatic variants are used.

6-    In table 1, it would be important to include the variant allele frequencies as well.

7-    Include software name used to generate figures 1 and 2A.

8-    Figure 4 should be labelled A,B,C,D, same for figure 5 and 6.

Author Response

1 Please include more information on variants allele frequency in the gnomAD and ExAC databases.

Answer: We added supplementary material_Table S1

2 Add more information on read depth and the total depth for calling variants at a specific locus.

Answer: We added supplementary material_Table S1

3 Details on filtering out synonymous mutations and allele frequency threshold for filtering non-synonymous mutations.

Answer: Separate filtering was not performed to allele frequencies for synonymous mutations and non-synonymous mutations.

4 How were raw variants filtered?

Answer: For Raw Variants, no separate filter was applied within the analysis result.

5 Are any of the variants present in the 1000 Genomes? This is required to make sure only somatic variants are used.

Answer: Without control samples, only gastric cancer cell line samples were analyzed, and individual Germline Variant Calls were made.

6 In table 1, it would be important to include the variant allele frequencies as well.

Answer: We added supplementary material_Table S1

7 Include software name used to generate figures 1 and 2A.

Answer: Excel used to generate Figure 1. Power point used to generate Figure 2A

8 Figure 4 should be labelled A,B,C,D, same for figure 5 and 6.

Answer: We labelled A,B,C,D in figure 4-6.

Round 2

Reviewer 2 Report

Thank you to the authors for incorporating the feedback provided. Recommend accepting the submission.